# WHAT MAKES FOR GOOD VISUAL TOKENIZER SUPERVISION FOR LARGE LANGUAGE MODELS?

## ABSTRACT

We empirically investigate proper pre-training supervision to build good visual tokenizers, making Large Language Models (LLMs) powerful Multimodal Large Language Models (MLLMs). In our benchmark, which is curated to evaluate MLLM's visual semantic understanding and fine-grained perception capabilities, we discussed different visual tokenizers pre-trained with dominant methods (*i.e.*, DeiT, CLIP, MAE, DINO and DINOv2), and observed that: i) Fully/weakly supervised models capture more semantics than self-supervised models, but the gap is narrowed by scaling up the pre-training dataset. ii) Self-supervised models are better at fine-grained perception, where patch-level supervision is particularly effective. iii) Tuning the visual tokenizer leads to the loss of semantics obtained from large-scale pretraining, which is unfavorable with the relatively small-scale instruction-tuning dataset. Given the findings, we reviewed methods that attempted to unify semantics and fine-grained visual understanding, *e.g.*, patch-level feature distillation with semantically-rich targets. We obtain an intriguing insight: *without further modification, mask-based strategies that were once all the rage may not be good visual tokenizer supervision*. Based on this critical observation, we obtain a new MLLM equipped with a tailored Good Visual Tokenizer – GVT, which exhibits strong visual comprehension capability at multiple scales. In particular, without introducing extra parameters and task-specific fine-tuning, GVT achieves superior performance on visual question answering, image captioning, and other fine-grained visual understanding tasks such as object counting and multi-class identification.

## 1 INTRODUCTION

Large Language Models (LLMs) (Brown et al., 2020; Touvron et al., 2023; Radford et al.; Ouyang et al., 2022) have demonstrated remarkable performance for various downstream tasks without task-specific fine-tuning. Recently, based on the powerful LLMs, there has been a surge of research (Li et al., 2023b; Alayrac et al., 2022; Zhu et al., 2023; Liu et al., 2023; Ye et al., 2023; Huang et al., 2023; Yang et al., 2023b; Driess et al., 2023) that successfully adapt LLMs to vision-language tasks, resulting in powerful Multimodal LLMs (MLLMs), *e.g.*, BLIP-2 (Li et al., 2023b). When properly fed with visual data, they are shown to be capable of understanding the visual world and responding to instructions accordingly. Such vision-language understanding capability makes LLM a universal interface for multimodal tasks, contributing towards a tentative yet promising direction towards Artificial General Intelligence (AGI) (Bubeck et al., 2023; OpenAI, 2023).

Within this framework, images are projected to the linguistic space for the LLMs to understand, where the common practice employs an image-text pre-trained visual tokenizer with contrastive supervision[1], *i.e.*, CLIP. However, even though CLIP has shown strong capacity for image representations, to the best of our knowledge, *it is yet to be explored whether CLIP is the optimal visual tokenizer for MLLMs*. The absence of such investigation calls for a comprehensive comparison of existing visual tokenizers under the MLLMs' framework. However, recent MLLMs have mostly investigated their performance in terms of generation quality (Zhu et al., 2023; Liu et al., 2023) or on a small set of questions (Ye et al., 2023), leaving an in-depth quantitative evaluation untouched.

---

[1]In this work, we study visual tokenizers that map images into a continuous latent space.

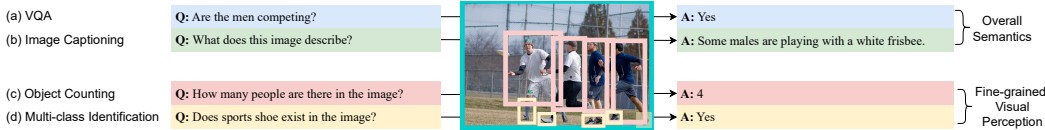

Figure 1: Different tasks require a visual understanding of different perspectives. Mainstream vision-language tasks (VQA and image captioning) mainly focus on the general and overall semantics of the image. In this work, to investigate the fine-grained visual understanding of a model, we also study two tasks: (c) Object Counting (OC) and (d) Multi-Class Identification (MCI), focusing on region and instance level understandings, respectively.

To this end, we curated a new benchmark to study what pretraining supervision makes for a Good Visual Tokenizer (GVTBench). It is specially designed to evaluate an MLLM's visual understanding capability from two important perspectives: semantic understanding and fine-grained visual perception capabilities. As shown in Figure 1, the former is evaluated on Visual Question Answering (VQA) and image captioning. While the latter is tested on two new tasks: Object Counting (OC) and Multi-Class Identification (MCI), which requires an in-depth understanding of fine-grained visual information. Based on this benchmark, we comprehensively evaluated existing visual tokenizers with identical architecture but different pretraining supervisions, including fully supervised DeiT (Touvron et al., 2021), text-guided weakly supervised CLIP (Radford et al., 2021) and self-supervised MAE (He et al., 2022), DINO (Caron et al., 2021), DINOv2 (Oquab et al., 2023) models (Section 2). Our main observations are **i)** fully supervised and text-guided weakly supervised visual tokenizers demonstrate better semantic representation capacity than their self-supervised counterparts, but the gap is narrowed by scaling up the pretraining dataset (*i.e.*, CLIP *vs.* DINOv2). **ii)** Self-supervised visual tokenizers show better fine-grained visual perception capacity, where patch-level supervision leads to superior region-level understanding. **iii)** On instruction tuning datasets which are often smaller than visual tokenizer pretraining dataset (Liu et al., 2023; Zhu et al., 2023), jointly tuning the visual tokenizer leads to noticeable semantic loss (*i.e.*, frozen CLIP performs much better than tunable CLIP on semantic understanding tasks).

Given the fact that none of the previous visual tokenizers exhibit both good semantic and fine-grained visual perceptual capabilities, we reviewed existing methods that integrate semantic and regional supervision and questioned whether they bring the best of the two worlds into a single visual tokenizer. Existing methods can be mainly divided into two categories. Methods in the first group (Zhong et al., 2022; Ye et al., 2023) enhance a pretrained CLIP with region-level supervision, which comes from a pretrained Region Proposal Network (RPN) or bounding box annotations. However, we found that this leads to the loss of original semantics, which can not be justified by the limited improvements in fine-grained visual perception capabilities. The other group of methods (Fang et al., 2023; Wei et al., 2022b) utilizes patch features from a pretrained CLIP as region supervision to train a new model, intending to enhance its fine-grained visual perceptual capability while maintaining the rich semantics. Specifically, Fang et al. (2023) and Wei et al. (2022a) use CLIP features to supervise the training of Masked-Image-Modeling (MIM), while Feature Distillation (Wei et al., 2022b) directly distills the CLIP feature into a new model without patch masking. Nonetheless, the introduction of `[MASK]` token in MIM leads to train-test mismatch, requiring the visual tokenizer to be jointly optimized in the instruction-tuning process, which again leads to semantic loss with the small-scale instruction tuning dataset. As such, we argue that, without architectural modification, the mask-based strategies that were once all the rage may not be good visual tokenizer supervision under MLLM's framework.

Based on these insights, we seek a new visual tokenizer with both strong semantic understanding and fine-grained visual perception capabilities via Feature Distillation (Wei et al., 2022b). Specifically, given a pretrained CLIP with rich semantics, we distill it into a new model by using the patch features as supervision, without patch masking. In this way, the rich semantics from large-scale image-text contrastive pretraining is preserved, and the fine-grained visual perceptual capability is greatly enhanced with patch supervision. With our new visual tokenizer and the language model Vicuna (FastChat, 2023), we obtain a new MLLM with **G**ood **V**isual **T**okenizer (GVT). Benefiting from the versatile visual tokenizer, GVT is able to perform well vision language tasks that require visual understanding at multiple levels. Without introducing extra parameters, we achieve superior

Table 1: Detailed Statistics of GVT-Bench.

| Task | Dataset | Evaluation Dimension | #Questions | Question Type | Answer Type |
|------|---------|---------------------|-----------|---------------|-------------|
| VQA | VQAv2 | General semantics | 440k | Multiple | Free-form Text |
| Image Captioning | MS-COCO | Overall semantics | 25k | What does the image describe? | Free-form Text |
| OC | MS-COCO&VCR | Region understanding | 20k | How many obj are there in the image? | Number |
| MCI | MS-COCO&VCR | Instance understanding | 20k | Does obj exist in the image? | Yes/No |

performance on semantic understanding tasks, *i.e.*, VQA and image captioning, as well as fine-grained visual understanding tasks: instance counting and multi-class identification.

To summarize, our contributions are as follows:

- To effectively evaluate MLLM's visual understanding capacity at different levels, we curate a new benchmark (GVTBench) which includes both semantic understanding tasks (VQA and image captioning) as well as fine-grained visual understanding tasks (Object Counting and Multi-Class Identification). Based on GVTBench, we perform extensive experiments to study what makes for good visual tokenizer supervision for MLLMs and make three main observations.

- We reviewed methods that combine CLIP with fine-grained supervision to see if they can achieve the best of both worlds in terms of visual semantics and fine-grained understanding. We found that the SOTA pre-trained models (*i.e.*, EVA) are inapplicable due to the train-test mismatch caused by MIM. Such mask-based visual tokenizers rely on further tuning with instructions, which leads to the loss of pre-trained rich semantics.

- Based on the insights, we tailor a new visual tokenizer by distilling the patch-level semantics of a pre-trained CLIP without masking. With our visual tokenizer and Vicuna, we arrive at a superior MLLM (GVT) with strong visual understanding capability, achieving state-of-the-art performance on our curated benchmark.

## 2 GVTBENCH FOR EMPIRICAL STUDY

To comprehensively study what makes for good visual tokenizer supervision for MLLMs, we conduct a series of experiments to study the properties of various visual tokenizers with the same architecture but different pretraining methods. In this work, we mainly investigate MLLMs' visual understanding capability from two important perspectives: semantic understanding and fine-grained visual perception.

### 2.1 EXPERIMENTAL SETUP

**GVTBench.** A comprehensive evaluation requires a benchmark that suitably quantifies MLLM's visual understanding capability. Nonetheless, existing vision-language tasks mainly focus on general and overall semantics Farhadi et al. (2010); Goyal et al. (2017), leaving a special focus on fine-grained visual perception untouched. To this end, we curated a new benchmark – GVTBench. It evaluates the semantic understanding capability of an MLLM on VQA (Goyal et al., 2017) and Image Captioning (IC) (Lin et al., 2014). We report accuracy for the former and CIDEr Vedantam et al. (2015) for the latter. For fine-grained visual perception capability evaluation, we specially designed two new tasks for MLLMs:

- **Object Counting (OC)**. We ask the model to count the number of certain objects appearing in the image with the prompt *"Question: How many {obj} are there in the image? Answer:"*. We regard it as a classification task and report a model's prediction accuracy.

- **Multi-Class Identification (MCI).** We ask the model if a certain object exists in the image with the prompt *"Question: Does {obj} exist in the image? Answer:"*. The model is expected to answer *" Yes/No"*, resulting in a binary classification problem. We report accuracy for this task.

Notably, in the VQAv2 (Goyal et al., 2017) benchmark, there are also questions related to numbers and small-scale objects. Nevertheless, these questions are of high diversity and are often coupled

with other semantic relations, making it unsuitable to strictly evaluate fine-grained visual understanding capabilities. For example, to answer a typical question "How many people are sitting on the bench?" in VQAv2, the model should first understand the relation (sit_on), which is thus not suitable for evaluating fine-grained visual understanding solely. In contrast, our OC and MCI tasks evaluate MLLM's understanding of individual objects, which is decoupled from semantic relations and thus a more appropriate test bed for fine-grained visual understanding evaluation.

To summarize, there are a total of 4 tasks in our GVTBench. (1) VQAv2 covers questions of various types. We thus take this benchmark to evaluate the general semantic understanding capability of a model. This task requires the model to have a good understanding of various high-level semantics in the image, including relatively abstract concepts such as actions and relations. (2) We use image captioning to quantify the capability of overall semantic understanding, which requires the model to understand the global information of the image. It requires the model to have a proper comprehension of the image and grasp the overall information such as the main activity and theme. Furthermore, we curated (3) OC and (4) MCI to evaluate MLLM's region-level and instance-level understanding capability, respectively. Compared to the former two tasks, the latter two tasks are totally decoupled from other semantics such as actions and relations, resulting in a better focus on fine-grained visual understanding. The details of GVTBench are shown in Table 1.

**Experimental Setting.** We use visual tokenizers with different supervision to encode an image into a set of visual tokens. Then, we follow Flamingo (ml_foundations, 2023) to use the Perceiver Resampler (Jaegle et al., 2021) to reduce the number of visual tokens to a fixed length, which are fed into LLM (*i.e.*, Vicuna). The models are trained on an instruction-tuning dataset which contains about 5M image-text pairs. In the training process, the language model is always frozen, while the visual tokenizer can be frozen or jointly optimized. More details are deferred to the appendix.

## 2.2 COMPARING VISUAL TOKENIZERS

On GVTBench, we evaluate visual tokenizers with the same architecture ViT-B (Dosovitskiy et al.) but different pretraining strategies, including fully supervised DeiT (Touvron et al., 2021), self-supervised DINO (Caron et al., 2021), DINOv2 (Oquab et al., 2023), MAE (He et al., 2022) and text-guided weakly supervised CLIP (Radford et al., 2021). To further investigate the effect of pre-training dataset size, we also compared a CLIP pretrained with 20M image-text pairs, using the checkpoint provided by (Yang et al., 2023a). Based on the results in Table 2, we arrive at the following observations.[2]

**Fully/weakly supervised models capture more semantics than self-supervised ones, but the gap is narrowed or even mitigated by scaling up the pre-training dataset.** With tokenizers pretrained on relative small-scale dataset (*i.e.*, ImageNet-1k (Russakovsky et al., 2015)) with 1.28M images), DeiT demonstrates better image captioning performance (65.8 CIDEr) than self-supervised models DINO (45.0) and MAE (37.3), without jointly tuning the visual tokenizer. However, with 142M images for pretraining, the self-supervised model – DINOv2 outperforms the supervised DeiT on image captioning (67.9) and VQA (51.3), and is only inferior to CLIP which is pretrained with weak supervision from a large-scale dataset with 400M image-text pairs.

**Self-supervised models are better at fine-grained perception, where patch-level supervision is particularly effective.** On fine-grained visual understanding tasks, *i.e.*, OC and MCI, self-supervised models demonstrate consistently better performance than those with supervision. When they are jointly tuned on the instruction dataset, their OC and MCI performance are mostly boosted, indicating their fine-grained visual perception capability gets improved. Among all the self-supervised models, MAE achieves the best performance, indicating the patch-based supervision is particularly effective for improving fine-grained visual understanding.

**Tuning semantic-rich visual tokenizer leads to semantic loss on small-scale instruction tuning dataset.** When the tokenizer is jointly optimized on the instruction tuning dataset, the rich semantics obtained from large-scale pretraining in CLIP and DINOv2 have noticeably dropped (*e.g.*, CLIP VQA $52.2 \rightarrow 47.7$ and DINOv2 captioning $67.9 \rightarrow 49.6$). We conjecture this is due to the relatively small scale of our instruction dataset ($\sim$5M $\ll$ 142M). As such, for modern MLLMs that are often

---

[2]Note these strategies adopt diverse protocols for pretraining, due to their inherent disparities. We thus adopt the off-the-shelf checkpoints for a fair comparison.

tuned on small-scale and high-quality instruction datasets (Zhu et al., 2023; Liu et al., 2023), jointly tuning the visual tokenizer may not be a good option.

Table 2: Comparison of visual tokenizers with different pretraining strategies. The **best** result is **bold** while the second best is underlined.

| Tuning | Supervision | Visual Tokenizer | # Images | VQA | COCO-Caption | COCO-OC | COCO-MCI | VCR-OC | VCR-MCI | Avg |
|---|---|---|---|---|---|---|---|---|---|---|
| ✕ | Fully | DeiT (Touvron et al., 2021) | 1.28M | 48.3 | 65.8 | 37.5 | 83.6 | 29.7 | 62.5 | 54.6 |
| | Self | DINO (Caron et al., 2021) | 1.28M | 50.1 | 45.0 | 46.5 | 80.8 | 33.1 | 56.3 | 52.0 |
| | | MAE (He et al., 2022) | 1.28M | 48.4 | 37.3 | **47.5** | 82.7 | 24.2 | 60.3 | 50.1 |
| | | DINOv2 (Oquab et al., 2023) | 142 M | 51.3 | 67.9 | 47.0 | 86.0 | 33.3 | 61.5 | 57.8 |
| | Weakly | CLIP-20M (Yang et al., 2023a) | 20 M | 48.2 | 60.9 | 42.5 | 79.1 | 26.5 | 58.3 | 52.6 |
| | | CLIP (Radford et al., 2021) | 400 M | **52.2** | **69.3** | 42.5 | 86.0 | 33.4 | 71.2 | **59.1** |
| ✓ | Fully | DeiT (Touvron et al., 2021) | 1.28M | 50.7 | 38.4 | 41.0 | 86.9 | 31.2 | 63.6 | 52.0 |
| | Self | DINO (Caron et al., 2021) | 1.28M | 47.3 | 54.1 | 44.5 | 86.6 | 30.2 | 57.3 | 53.3 |
| | | MAE (He et al., 2022) | 1.28M | 48.9 | 48.0 | **47.5** | **88.7** | **34.8** | 71.4 | 56.7 |
| | | DINOv2 (Oquab et al., 2023) | 142 M | 50.5 | 49.6 | 43.5 | 84.1 | 33.2 | 68.9 | 55.0 |
| | Weakly | CLIP-20M | 20 M | 49.6 | 61.2 | 37.0 | 84.5 | 30.0 | 62.2 | 54.1 |
| | | CLIP (Radford et al., 2021) | 400 M | 47.7 | 64.2 | 45.5 | 88.0 | 34.5 | 68.8 | 58.1 |

# 3 Unifying Semantic and Fine-grained Visual Understanding

## 3.1 CLIP with Region-based Training

The generalist MLLMs call for a versatile visual tokenizer that could properly represent an image's content at multiple levels. However, based on the results in Table 2, none of existing pretraining methods leads to a good visual tokenizer that excels at both semantic and fine-grained visual perception capabilities. This motivates us to explore whether the best of the two worlds can be achieved by any other method.

**Fine-tuning CLIP with region supervision.** One stream of work (Zhong et al., 2022; Minderer et al., 2022) attempted to improve region representation capability of a pretrained CLIP by fine-tuning it with region supervision, which has demonstrated improved performance for open-vocabulary object detection. This motivates us to study if this also enhances CLIP as a visual tokenizer. We mainly investigated RegionCLIP (Zhong et al., 2022) and Owl-ViT (Minderer et al., 2022). The former finetune a CLIP with region-level supervision from bounding boxes generated by a pretrained RPN, while the latter utilizes the region annotation from an object detection dataset. We compared these methods with CLIP, and show the results in Table 3. It can be observed that without joint tuning the visual tokenizer, both RegionCLIP and Owl-ViT show severe performance drop on image captioning and VQA, indicating the rich semantics in the original CLIP is lost during their region fine-tuning process. On the other hand, when the visual tokenizers are jointly tuned on the instruction-tuning dataset, their fine-grained representation capability improves by a margin (on OC and MCI performance), but this can not justify the loss of semantic representation capability, resulting in inferior overall performance compared to the original CLIP.

Table 3: Comparing CLIP with its region-tuned counterparts.

| Tuning | Visual Tokenizers | VQAv2 | COCO-Caption | COCO-OC | COCO-MCI | VCR-OC | VCR-MCI | Avg |
|---|---|---|---|---|---|---|---|---|
| ✕ | CLIP (Radford et al., 2021) | **52.2** | **69.3** | 42.5 | 86.0 | 33.4 | 71.2 | **59.1** |
| ✕ | RegionCLIP (Zhong et al., 2022) | 48.7 | 28.5 | 41.0 | 86.0 | 34.1 | 70.9 | 51.5 |
| ✕ | Owl-ViT (Minderer et al., 2022) | 44.0 | 32.5 | 43.0 | 80.8 | 33.5 | 68.3 | 50.4 |
| ✓ | CLIP (Radford et al., 2021) | 47.7 | 64.2 | 45.5 | **88.0** | **34.5** | 68.8 | 58.1 |
| ✓ | RegionCLIP (Zhong et al., 2022) | 49.7 | 65.5 | **47.5** | 86.4 | 34.1 | 69.1 | 58.7 |
| ✓ | Owl-ViT (Minderer et al., 2022) | 50.8 | 61.2 | 38.5 | 87.1 | 34.2 | **71.3** | 57.2 |

**Semantic Feature as Region Supervision.** Another stream of work utilized CLIP's patch feature as region-level supervision for pretraining, aiming to obtain a model with both strong semantics and better region representations. Specifically, EVA (Fang et al., 2023) and MVP (Fang et al., 2023) use CLIP's patch feature as regression target for Masked Image Modeling (MIM) pretraining, while FD (Wei et al., 2022b) does not employ the masking strategy and directly distills CLIP's patch feature into a new model. We compared these methods in Table 4. Without jointly tuning the visual tokenizer, FD results in performance improvement on both semantic and fine-grained visual understanding upon CLIP. However, when a patch masking strategy is adopted, the performance of EVA significantly drops. This can be attributed to the introduction of the [MASK] token for MIM,

which is only used for pretraining the visual tokenizer but discarded afterward. In this way, the train-test mismatch arises without tuning the visual tokenizer, leading to unsatisfactory performance for downstream tasks. On the other hand, when the visual tokenizer is jointly optimized with the instruction data, they are inferior to the original CLIP on VQA and image captioning, indicating semantic loss occurs.

Given the fact that modern MLLMs are often trained on high-quality and small-scale instruction datasets (Zhu et al., 2023; Liu et al., 2023), our observation suggests that visual tokenizer should be frozen to maintain the powerful semantic representation capability from large-scale pretraining. Nonetheless, for visual tokenizers pretrained with MIM, the introduction of the [MASK] token inevitably leads to train-test mismatch, necessitating it to be jointly tuned on the instruction data. This contradiction indicates that mask-based pretraining may not lead to a good visual tokenizer under MLLM's framework.

As such, even though the results in Table 2 suggest that region-level supervision is effective for fine-grained visual understanding, it should be carefully utilized under the MLLMs framework. To properly utilize it to improve CLIP's fine-grained visual perceptual capabilities, the results in Table 4 demonstrate that, with its current architecture, the mask-based strategies that were once all the rage may not lead to good visual tokenizer supervision.

Table 4: Comparison of different strategies that utilize CLIP features as region supervision.

| Method | Tuning | Mask | VQAv2 | COCO-Caption | COCO-OC | COCO-MCI | VCR-OC | VCR-MCI | Avg |
|---|---|---|---|---|---|---|---|---|---|
| CLIP (Radford et al., 2021) | × | - | **52.2** | 69.3 | 42.5 | 86.0 | 33.4 | 71.2 | 59.1 |
| FD (Wei et al., 2022b) | × | × | 49.4 | **72.1** | 46.5 | 86.7 | 34.2 | **72.3** | **60.2** |
| EVA (Fang et al., 2023) | × | ✓ | 42.9 | 27.0 | **46.9** | 70.5 | 21.6 | 59.9 | 44.8 |
| CLIP (Radford et al., 2021) | ✓ | - | 47.7 | 64.2 | 45.5 | **88.0** | **34.5** | 68.8 | 58.1 |
| FD (Wei et al., 2022b) | ✓ | × | 49.3 | 53.3 | 40.5 | 85.8 | 32.1 | 70.2 | 55.2 |
| EVA (Fang et al., 2023) | ✓ | ✓ | 51.4 | 61.6 | 45.9 | 87.1 | 31.4 | 69.8 | 57.9 |

## 3.2 MLLM WITH GOOD VISUAL TOKENIZER

Based on the insights above, we found the patch supervision introduced by feature distillation is helpful in maintaining the semantic representation capability of CLIP while improving its fine-grained perceptual capabilities. As such, we tune a new visual tokenizer that unifies the advantages of semantic representation and fine-grained visual perception capabilities. In particular, we achieve this objective by utilizing a visual tokenizer pretrained on large-scale datasets and properly integrating it with patch-level supervision. Motivated the findings in Table 4, we do not use any mask-based strategy, so the rich semantics could be preserved by freezing it in the instruction tuning process. To achieve stronger performance, we take the powerful EVA-CLIP (Sun et al., 2023) based on ViT-L as the teacher model and randomly initialize another model with identical architecture as the student. During training, each image is fed into the teacher and student model, obtaining the representation $\mathbf{t}$ and $\mathbf{s} \in \mathbb{R}^D$ for each image patch, respectively. Then, we perform feature distillation with the following objective:

$$\mathcal{L}_{\text{distill}}(\mathbf{s}, \mathbf{t}) = \begin{cases} \frac{1}{2}\left(g(\mathbf{s}) - \text{whiten}(\mathbf{t})\right)^2/\beta, & \text{if } |g(\mathbf{s}) - \text{whiten}(\mathbf{t})| \leq \beta \\ |g(\mathbf{s}) - \text{whiten}(\mathbf{t})| - \frac{1}{2}\beta, & \text{otherwise} \end{cases} \quad (1)$$

The patch features from the student model are first passed through a learnable function $g(\cdot)$, which is a $1\times1$ convolution layer. The whitening operation is utilized to stabilize the training process, which is implemented as a non-parametric layer normalization without scaling and bias Wei et al. (2022b). In the FD process, only the student model and the projector $g(\cdot)$ are used for training, while the teacher model is frozen.

Based on the tuned visual tokenizer, we construct a new MLLM with **G**ood **V**isual **T**okenizer (GVT). The framework of GVT is shown in Figure 2. Following (ml_foundations, 2023), we also random initialize a Receiver Resampler (Jaegle et al., 2021) with 32 learnable queries to attend to the features from the visual tokenizer. Then, the features from the Perceiver Resampler are fed into the LLM (Vicuna-7B (FastChat, 2023))together with the language prompts. The whole model is trained by the language modeling loss, and only the Perceiver Resampler is optimized in this process.

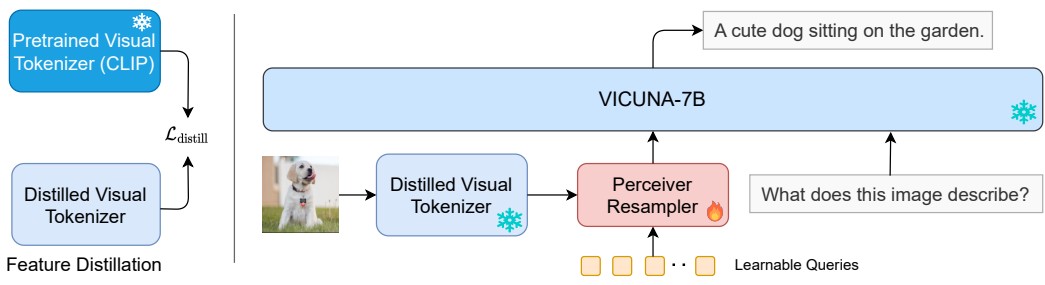

Figure 2: The framework of our GVT. We first distill the features of a pretrained CLIP via smoothed $\mathcal{L}_1$ loss. Then, we use it to encode images into a set of tokens, which are fed into the Perceiver Resampler (Jaegle et al., 2021) as soft prompts. Together with language instructions, these prompts are fed into LLM to generate responses. Only the Perceiver Resampler is optimized in this process.

## 4 EXPERIMENTS

### 4.1 EXPERIMENTAL SETUP

We train our model on a joint dataset of image-text pairs, including CC3M (Sharma et al., 2018), SBU (Vicente et al., 2016), Visual Genome (Krishna et al., 2017) and MS-COCO (Lin et al., 2014). We formulate these datasets as image captioning task, and use *"what does the image describe?"* as prompt during training. Besides, we also use two object detection datasets – Object365 (Shao et al., 2019) and OpenImagesV6 (Kuznetsova et al., 2020) to design a set of object-centric tasks following (Piergiovanni et al., 2022). The LLaVA-150k (Liu et al., 2023) dataset is also utilized for joint training. This results in a total of 15M image-text pairs. The images are resized to 224 × 224, and we adopt random resized crop and horizontal flipping for data augmentation during training. The model is trained for 50k steps with 2k steps for linear warmup. We use AdamW ((Loshchilov & Hutter, 2017)) optimizer with a learning rate of 1e-4 and batch size 1024. The training process takes about 2 days on 32 Tesla V100 GPUs. For feature distillation, we followed the training protocol in Wei et al. (2022b) except that we trained the model for a total of 50 epochs on the ImageNet-1k (Russakovsky et al., 2015) dataset due to its high quality. The $\beta$ is set to 2.0 through the process. For more implementation details, please refer to our appendix.

### 4.2 COMPARISON WITH OTHER MLLMs

We evaluate GVT on our GVTBench, and compare it with recent MLLMs, including Flamingo (ml_foundations, 2023), KosMos-1 (Huang et al., 2023), BLIP-2 (Li et al., 2023b), , LLaVa (Liu et al., 2023), MiniGPT4 (Zhu et al., 2023). The results are shown in Table 5.

Our GVT achieves the best overall performance across competitors. Specifically, on tasks requiring fine-grained visual perception, *i.e.*, OC and MCI on both COCO and VCR datasets, GVT surpasses models with larger visual tokenizer and more curated data. This indicates our visual tokenizer can better capture the fine-grained visual information, providing representations with better details. For semantic understanding tasks including VQA and image captioning, GVT achieves the second-best result. It is only inferior to BLIP-2, which utilized a much larger instruction dataset with high-quality image captions filtered by (Li et al., 2022).

Table 5: Comparison with MLLMs. The **best** results are bold and the second best is underlined.

| Model | #Vis. Tok. Size | VQAv2 | COCO-Caption | COCO-OC | COCO-MCI | VCR-OC | VCR-MCI | Avg |
|---|---|---|---|---|---|---|---|---|
| Flamingo-9B (Alayrac et al., 2022) | 438 M | 51.8 | 79.4 | - | - | - | - | - |
| Kosmos-1 (Huang et al., 2023) | 307 M | 51.0 | 84.7 | - | - | - | - | - |
| LLaVa (Liu et al., 2023) | 307 M | 39.0 | 48.3 | 22.2 | 52.0 | 24.6 | 66.9 | 44.7 |
| MiniGPT4 (Zhu et al., 2023) | 1.0 B | 58.2 | 80.6 | 21.5 | 76.8 | 25.1 | 70.1 | 55.4 |
| BLIP-2 (Li et al., 2023b) | 1.0 B | **62.4** | **93.3** | 48.0 | 81.9 | 20.2 | 68.9 | 62.5 |
| GVT (Ours) | 307 M | 60.4 | 89.9 | **56.2** | **89.3** | **40.3** | **78.9** | **69.2** |

## 4.3 ABLATION STUDY

**Effect of Feature Distillation.** To further validate the effectiveness of Feature Distillation, we compared the visual tokenizer before and after in Table 6. It can be observed that the distilled visual tokenizer achieves comparable performance on semantic understanding tasks (VQA and Image Captioning), while greatly improving fine-grained visual perception tasks (OC and MCI), resulting in improved overall performance. This observation is aligned with our findings in Section 3, where feature distillation consistently improves model performance across different architectures. We also provide an evaluation on SEED-Bench Li et al. (2023a), which is a recently released MLLM benchmark focusing on visual understanding. In Table 7, FD improves performance more on fine-grained understanding tasks such as instance identity, location, and counting.

Table 6: Comparison between visual tokenizer with and without FD.

| Visual | VQAv2 | COCO-Caption | COCO-OC | COCO-MCI | VCR-OC | VCR-MCI | Avg |
|---|---|---|---|---|---|---|---|
| EVA-CLIP | **60.5** | **90.8** | 43.5 | 85.6 | 37.6 | 71.1 | 64.9 |
| EVA-CLIP-FD | 60.4 | 89.9 | **56.2** | **89.3** | **40.3** | **78.9** | **69.2** |

Table 7: Comparison between with and without FD on SEED-Bench Li et al. (2023a).

| Visual | Scene | Inst.Id | Inst.Loc | Inst.Attr | Inst.Count | Spatial | Interaction | Reason | Avg |
|---|---|---|---|---|---|---|---|---|---|
| EVA-CLIP | 41.26 | 34.30 | 31.40 | **29.84** | 34.81 | **32.98** | **31.96** | 50.75 | 35.93 |
| EVA-CLIP-FD | **41.74** | **35.50** | **31.79** | 29.45 | **36.17** | 31.96 | **31.96** | **51.06** | **36.20** |

Table 8: Comparison of visual tokenizer with different LLMs.

| LLM | Visual Tokenizer | VQAv2 | COCO-Caption | COCO-OC | COCO-MCI | VCR-OC | VCR-MCI | Avg |
|---|---|---|---|---|---|---|---|---|
| Flant5-xxl | EVA-CLIP | **55.8** | **68.1** | 42.5 | 70.6 | 19.9 | 66.6 | 53.9 |
| | EVA-CLIP-FD | 55.4 | 67.2 | **43.6** | **71.4** | **20.3** | **66.8** | **54.1** |
| LLaMa-7B | EVA-CLIP | **54.2** | 66.3 | 42.9 | 68.3 | 17.3 | 54.4 | 50.6 |
| | EVA-CLIP-FD | 53.9 | **67.5** | **43.2** | **70.3** | **18.9** | **56.2** | **51.7** |

**Effectiveness with Different LLMs.** Our GVT is trained with our distilled visual tokenizer and Vicuna-7B as LLM. In fact, our distilled visual tokenizer is also effective with different LLMs. In Table 8, our distilled visual tokenizer can generally improve the overall performance when using Flant5-xxl and LLaMa-7B as LLM, with the performance on OC and MCI particularly improved.

## 4.4 VISUALIZATIONS

**Attention Maps.** To further understand how FD improves fine-grained understanding, we selected one query in the perceiver resampler, and visualized the attentions in two heads in Figure 3. It can be observed that, without FD, the attention mostly focuses on the salient areas of the image, and the attention maps in two different heads are generally similar. In contrast, with FD, the attention maps exhibit higher diversity, which is aligned with Wei et al. (2022b). Also, the attention may focus more on informative but non-salient regions (*e.g.*, broccoli and bike in the last column).

**Qualitative Results.** We show some qualitative comparison of OC and MCI between our GVT and BLIP-2 in Figure 4. It can be observed that our method demonstrates better fine-grained visual understanding capabilities than the baseline method. Take the first example in OC as an example, our method not only recognizes the 3 people in the foreground but also takes the fourth person who is far away from the camera into consideration. Besides, GVT also successfully recognizes non-salient or small-sized objects in the image, such as the bicycle and broccoli in MCI.

## 5 RELATED WORK

**Multimodal Large Language Models.** Recently, with the open source of Large Language Models (Touvron et al., 2023; FastChat, 2023; Radford et al.; Chung et al., 2022), a lot of large multimodal models are constructed based on them. Mini-GPT4 (Zhu et al., 2023) is built on the instruction-tuned Vicuna (FastChat, 2023) and the visual encoder from BLIP-2 (Li et al., 2023b), with only a linear layer trained to bridge the two modules. This simple design results in a powerful multi-modal chatbot, with noticeable vision-language understanding capability. LLaVa (Liu et al., 2023) adopts CLIP as visual tokenizer, and trains the projector with a curated dataset with balanced concepts. The model then can be finetuned for downstream tasks, *e.g.*, ScienceQA (Lu et al., 2022). Apart from using frozen visual tokenizer, mPLUG-OWL (Ye et al., 2023) tunes the Perceiver Resampler with large-scale image-text data in the first stage, followed by the finetuning of the language model with LoRA (Hu et al.) in the second stage. Although these generalist models have demonstrated impressive capability on multimodal tasks, we find that they mostly focus on the general or overall semantic understanding of the image, ignoring more fine-grained visual perception.

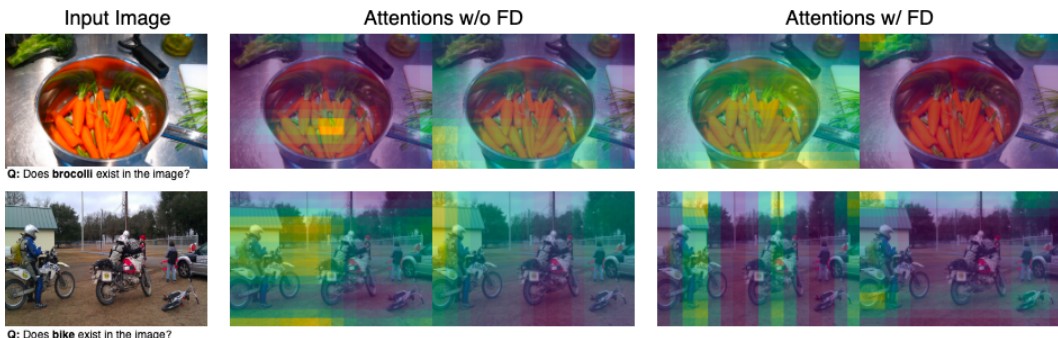

Figure 3: Visualization of feature maps with and without FD. We select one query in the perceiver resampler and visualize its attention map in two fixed heads over the image. With FD, the attention maps not only show higher diversity but also focus more on informative and fine-grained regions (*e.g.*, broccoli in the first row and bike in the second row).

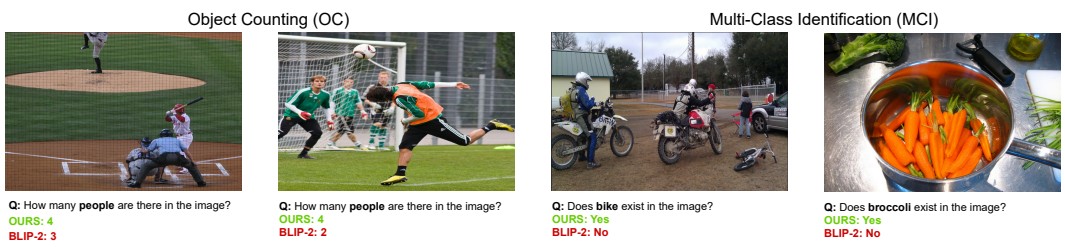

Figure 4: Qualitative comparison on OC and MCI. Our model is better at fine-grained perception.

**Visual Tokenizer Pretraining.** Visual encoders have been shown to benefit from large-scale pre-training for downstream tasks. The most common approach first pretrains the model on a large dataset with annotations, *e.g.*, ImageNet (Russakovsky et al., 2015), and finetunes it for downstream tasks such as semantic segmentation (Zhou et al., 2019) and object detection (Lin et al., 2014). Recently, self-supervised pre-training have also shown to improve model's representation capability. The typical contrastive-based methods (Caron et al., 2021; Chen et al., 2020; Chen & He, 2021) trains the model by aligning views from the same image. Inspired by the idea of mask-language-modeling for pretraining language models (Kenton & Toutanova, 2019), masked-image-modeling has also evolved for visual encoder pretraining. These methods mask a proportion of image patches before feeding them into the model, and ask the model to recover the masked patches. Some methods (Bao et al.) discretize the masked patches via a pretrained tokenizer (Ramesh et al., 2021). Recently, auto-encoder based (He et al., 2022) methods ask the model to directly generate the masked patch in the continuous space. Another stream of visual encoders is pretrained on massive image-text pairs via contrastive learning (Radford et al., 2021), achieving strong zero-shot understanding.

## 6 CONCLUSION AND FUTURE WORK

We comprehensively studied various visual tokenizer supervisions through the lens of MLLM. Our investigation reveals that i) fully/weakly supervised models perform generally better than self-supervised ones on semantic representation. ii) Self-supervised models are better at fine-grained visual perception, where patch-level supervision is particularly effective. iii) jointly tuning the visual tokenizer on the small-scale instruction dataset leads to the loss of rich semantics from large-scale pretraining. Then, we seek a visual tokenizer supervision that excels at both semantic understanding and fine-grained visual perception. We reviewed existing methods and found that directly fine-tuning CLIP with region supervision does not lead to a versatile visual tokenizer. Besides, the masking strategy for pretraining is not suitable due to the train-test mismatch. Based on the insights above, we tune a new visual tokenizer, which distills the CLIP patch feature into a new model without masking. With our visual tokenizer, Vicuna can better understand images at multiple levels, resulting in superior performance on various vision-language tasks. For future work, we would like to explore a more versatile visual tokenizer that can handle more challenging visual understandings.

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
