# WHAT MAKES FOR GOOD VISUAL TOKENIZERS FOR LARGE LANGUAGE MODELS? – APPENDIX

## 1 IMPLEMENTATION DETAIL

### 1.1 IMPLEMENTATION DETAIL FOR EMPIRICAL STUDIES

For the experiments in empirical studies, we use a combination of 1) image captioning datasets: MS-COCO (Lin et al., 2014), SBU (Vicente et al., 2016), CC-3M (Sharma et al., 2018) and Visual Genome (Krishna et al., 2017) and 2) two object detection datasets, including Object365 (Shao et al., 2019) and OpenImagesV6 (Kuznetsova et al., 2020). For image captioning data, we take the question *"what does the image describe?"* as input prompt and ask the model to generate the descriptions. For object detection datasets, we use a total of 6 tasks to fully utilize the rich annotations. Please refer to Section D in this appendix for more details. The training dataset is uniformly sampled during training. We optimize the model with a learning rate of 1e-4 and a batch size 1024. The whole model is optimized by the AdamW (Loshchilov & Hutter, 2017) optimizer and we set $\beta_1$ to 0.9 and $\beta_2$ to 0.98. We train the model for 10k steps, while the learning rate is linearly warmuped from 0 in the first 1k steps, and is cosine decayed to 0 afterwards. We optimize all models using `float16`.

### 1.2 IMPLEMENTATION DETAIL FOR GVT

The implementation detail of our GVT is similar to that in the empirical studies, except that we use more data and more training steps. Besides the image captioning and object detection dataset, we also used LLaVa-158k dataset (Liu et al., 2023), which is generated by external powerful LLM. We trained the model for 50k steps, with 2k steps for linear warmup. Then, we use cosine decay to decrease the learning rate to 0.

### 1.3 EVALUATION DETAILS.

**VQA.** Modern Language Models mainly generate one or multiple sentences, making it infeasible to directly evaluate the MLLMs in the standard evaluation protocol which requires the prediction and ground truth to be exactly matched. As such, we slightly relax the original evaluation protocol. We use the first sentence generated my MLLM as prediction result, and treated it as correct if *contains* the ground truth answer.

**Image Captioning.** When MLLMs generate multiple sentences, we use the first sentence as the captioning result for evaluation. Since MLLMs tend to generate multiple sentences, we use the prompt *"Describe this image in a sentence: This is an image of"* as prompt to condense the prediction for effective evaluation.

**Object Counting.** We extract the number of word from the first generated sentence, and compare it with ground truth number.

**Object Existence.** We extract "yes" or "no" from the first generated sentence, and compare it with ground truth.

## 2 BENCHMARKING FINE-GRAINED VISUAL UNDERSTANDING TASKS

We provide the details of the dataset used for evaluation in each task in Table 1. In this work, we constructed two fine-grained perception tasks: *object counting* and *object existence* based on instance-level annotations from existing datasets. Specifically, they are constructed on MS-COCO (Lin et al., 2014) and VCR (Zellers et al., 2019) validation datasets. We provide their details as follows.

Table 1: Dataset set statistics of our dataset for evaluation.

| Task | Split | Dataset | # of Instance |
|---|---|---|---|
| Visual Question Answering | validation | VQAv2 (Goyal et al., 2017) | 440k |
| Image Captioning | validation | MS-COCO (Lin et al., 2014) | 25k |
| Object Counting | validation | MS-COCO (Lin et al., 2014) | 10k |
| Object Counting | validation | VCR (Zellers et al., 2019) | 10k |
| Multi-Class Identification | validation | MS-COCO (Lin et al., 2014) | 10k |
| Multi-Class Identification | valitdaion | VCR (Zellers et al., 2019) | 10k |

## 2.1 OBJECT COUNTING

Besides the visual features, the prompt of this task – *"Question: How many {obj} are there in the image? Answer:¨* is fed into the MLLM for evaluation. We select the object name *{obj}* from the object list of the dataset. Since there are often a single object of a certain class in one image, we select a maximum of 3 objects with highest occurrence in the image to make this benchmark challenging. Similar to object counting benchmarks, we report Mean Absolute Error (MAE) and Root Mean Square Error (RMSE). Furthermore, we also report accuracy which treats the counting as a classification problem during evaluation. Both COCO-OC and VCR-OC contain a total of 10k tasks.

## 2.2 MULTI-CLASS IDENTIFICATION

Multi-label classification can be used as task to evaluate the model's multi-instance understanding capability. However, given the open-ended nature of language models, the evaluation process is not stable since the language model may generate more fine-grained object names than the dataset categories, making a stable and fair evaluation difficult. To this end, we change the format of this task and make the evaluation process more stable. We design the prompt as *"Question: Does {obj} exist in the image?" Answer:¨*, and the model is expected to answer *"Yes"* or *"No"*. We select the object name *{obj}* from the object list of the dataset. For each image, we randomly select at most 3 objects that exist in the image, and the same number of objects that does not appear in the image, so as to make the evaluation set balanced. Both COCO-MCI and VCR-MCI contain a total of 10k tasks.

# 3 MORE FINE-GRAINED VISUAL UNDERSTANDING RESULTS

In this section, we provide more detailed results on our two new tasks: OC and OCI.

**Detailed Object Counting Results.** We show the detailed results of Object Counting task on MS-COCO in Table 2. It can be observed that, when the images contains relatively small number of objects (1-3), all methods can understand the number of objects to some extend, where ours is significantly better than others. However, when the images become more complex, where the number of occurrence increases (4-6, 7-9), the performance has significantly dropped. Similar trend can also observed in Table 3. These results demonstrate that current MLLMs still struggle at correctly counting the objects, indicating future research are required to make them more capable of challenging visual understanding tasks.

Table 2: Detailed results on the Object Counting on MS-COCO dataset.

| GT range | 1 - 3 | | | 4 - 6 | | | 7 - 9 | | | Overall | | |
|---|---|---|---|---|---|---|---|---|---|---|---|---|
| Method | Acc ↑ | MAE ↓ | RMSE ↓ | Acc ↑ | MAE ↓ | RMSE ↓ | Acc ↑ | MAE ↓ | RMSE ↓ | Acc ↑ | MAE ↓ | RMSE ↓ |
| MiniGPT4 | 23.0 | 0.96 | 1.60 | 11.0 | 1.68 | 2.19 | 0.0 | 4.09 | 4.24 | 21.1 | 1.36 | 2.1 |
| LLaVa | 26.5 | 0.89 | 1.86 | 11.0 | 1.72 | 3.25 | 1.58 | 4.75 | 5.83 | 22.0 | 1.36 | 2.70 |
| BLIP-2 | 61.1 | 0.47 | 0.82 | 12.1 | 2.10 | 2.50 | 0.47 | 4.97 | 2.57 | 48.0 | 1.15 | 2.05 |
| GVT (Ours) | 74.7 | 0.25 | 0.51 | 4.7 | 2.26 | 2.49 | 0.02 | 2.29 | 5.25 | 56.0 | 1.01 | 1.93 |

**Detailed Multi-Class Identification Results.** We provide more detailed results on MCI task for MS-COCO in Table 4. The performance of all methods decrease when the image becomes more

Table 3: Detailed results on the Object Counting on VCR dataset.

| GT range | 1 - 3 | | | 4 - 6 | | | 7 - 9 | | | Overall | | |
|---|---|---|---|---|---|---|---|---|---|---|---|---|
| Method | Acc ↑ | MAE ↓ | RMSE ↓ | Acc ↑ | MAE ↓ | RMSE ↓ | Acc ↑ | MAE ↓ | RMSE ↓ | Acc ↑ | MAE ↓ | RMSE ↓ |
| MiniGPT4 | 25.0 | 0.84 | 1.32 | 13.0 | 1.48 | 1.82 | 0.00 | 4.34 | 4.46 | 25.0 | 1.51 | 2.24 |
| LLaVa | 24.0 | 0.91 | 2.24 | 13.3 | 1.53 | 1.99 | 1.16 | 4.46 | 4.75 | 24.0 | 1.58 | 2.77 |
| GVT (Ours) | 63.9 | 0.36 | 0.61 | 5.94 | 2.22 | 2.46 | 0.00 | 4.96 | 5.18 | 40.0 | 1.49 | 2.41 |

complex (with more objects in the image). However, the results on the VCR dataset does not show a stable trend. We conjecture this can be related to the difference on the instruction tuning datasets, which leads the model to focus on different types of objects.

Table 4: Detailed results on the Multi-Class Identification on MS-COCO dataset.

| #Objects | 1 - 9 | 10 - 20 | > 20 | Overall |
|---|---|---|---|---|
| MiniGPT4 | 80.7 | 72.3 | 96.1 | 76.8 |
| LLaVa | 52.1 | 52.0 | 51.7 | 52.0 |
| BLIP-2 | 85.4 | 77.6 | 75.2 | 81.9 |
| GVT (Ours) | 89.7 | 87.0 | 84.5 | 88.2 |

Table 5: Detailed results on the Multi-Class Identification on VCR dataset.

| GT range | 1 - 9 | 10 - 20 | > 20 | Overall |
|---|---|---|---|---|
| MiniGPT4 | 71.2 | 70.2 | 71.1 | 70.8 |
| LLaVa | 67.1 | 66.6 | 66.8 | 66.9 |
| BLIP-2 | 67.6 | 70.3 | 70. | 68.9 |
| GVT (Ours) | 77.1 | 80.6 | 81.5 | 78.8 |

## 4 OBJECT-CENTRIC TASKS

The work of (Piergiovanni et al., 2022) has proposed 4 tasks to utilize object detection dataset for vision-language pretraining, including:

**1. List Objects**
Input: *"List all objects"*
Output: *"{obj1}, {obj2}, ..."*

**2. Object Existence**
Input: *"Does {obj} exist in the image?"*
Output: *"Yes/No."*

**3. Group Existence**
Input: *"Does all of {obj1}, {obj2} and {obj3} exists in the image?"*
Output: *"Yes/No."*

**4. Existence Selection**
Input: *"Which of {obj1}, {obj2}, {obj3} exist in the image?"*
Output: *"{obj1/2/3}"*

To further utilize the rich annotations in object detection datasets, we also design two tasks that facilitate the model's learning on fine-grained visual information.

**5. Object Counting**
Input: *"How many {obj}s are there in the image?"*
Output: *1-9.*

**6. Spatial Relation**
Input: *"What is the spatial relation between {obj1} and {obj2}? Choose one from Top/Top Left/Left/Bottom Left/Bottom/Bottom Right/Right/Top Right"*
Output: "*Top/Top Left/Left/Bottom Left/Bottom/Bottom Right/Right/Top Right*"

Task 6 is only performed when the selected *{obj1}* and *{obj2}* are unique in the image, so as to avoid the referring ambiguity problem. For all tasks, we use the input text as the prompt and ask the model to generate the output text. The loss is only computed on the output texts. For each image, the task is uniformly sampled on the two object detection datasets (Shao et al., 2019; Kuznetsova et al., 2020).

# 5 MORE ABLATION STUDIES

**Choice of Distillation Target.** According to the results in Table 1 in our main paper, we observe that DINOv2, which is pre-trained with self-supervision on a dataset with 142M images also demonstrates good overall performance. To find the best target for feature distillation, we compared it with the CLIP model from (Sun et al., 2023), both in ViT-L architecture. The results are shown in Table 6. It can be seen that CLIP has demonstrated better overall performance, which can be attributed to its large-scale pretraining dataset and advanced training strategies.

Table 6: Comparison of visual tokenizers under ViT-L architecture.

| Visual Tokenizer | VQA Acc | Captioning CIDEr | Captioning SPICE | COCO-OC Acc | COCO-MCI Acc | Avg |
|---|---|---|---|---|---|---|
| DINO-v2-Large | 53.9 | 69.9 | 15.0 | **45.5** | **83.6** | 63.2 |
| CLIP-Large | **55.5** | **71.9** | **16.5** | 45.2 | 83.5 | **64.0** |

**Number of Latent Queries.** We study the number of latent queries in the Perceiver Resampler. The results are shown in Table 7. It can be observed that the overall performance generally increases with the number of latent queries, where 32 query results in a satisfactory performance. Besides, increasing the number of queries to 64 leads to limited improvements.

Table 7: Comparison of the number of latent queries in the Perceiver Resampler.

| #Latent Query | VQA Acc | Captioning CIDEr | Captioning SPICE | COCO-OC Acc | COCO-MCI Acc | Avg |
|---|---|---|---|---|---|---|
| 8 | 53.4 | 60.0 | 15.4 | 50.0 | 78.0 | 60.3 |
| 16 | 55.0 | 61.7 | 15.8 | **51.1** | 83.5 | 62.8 |
| 32 | **55.5** | **71.9** | **16.5** | 45.2 | 83.5 | 64.0 |
| 64 | 54.0 | 71.1 | 16.4 | 47.0 | **84.2** | **64.1** |

**Training with different resolutions.** In our MLLM, the model use images with $256 \times 256$ resolution for training and inference. It is expected that increasing the image resolution can improve the model's performance on downstream tasks, especially on fine-grained visual understanding tasks. As such, we compared models trained with images of $512 \times 512$ resolution. The results are shown in Table 8. It can be observed that higher resolution indeed leads to better performance.

Table 8: Comparison of visual tokenizer with different LLMs.

| Resolution | VQAv2 | COCO-Caption | COCO-OC | COCO-MCI | VCR-OC | VCR-MCI | Avg |
|---|---|---|---|---|---|---|---|
| 256 | 53.0 | 65.9 | 40.3 | 67.4 | 21.3 | 67.4 | 52.3 |
| 512 | 55.1 | 70.2 | 43.5 | 73.9 | 24.2 | 71.0 | 56.3 |