# OpenReview forum: "What Makes for Good Visual Tokenizers for Large Language Models"
_ICLR.cc/2024/Conference — Submitted to ICLR 2024_

### Official Review · Reviewer_NrqM · 2023-10-28

**Soundness:** 3 good
**Presentation:** 3 good
**Contribution:** 2 fair
**Rating:** 6
**Confidence:** 3

**Summary:**

This paper investigates how to improve the visual encoders for Multimodal Large Language Models (MLLM).

* The authors start by curating a more comprehensive benchmark for evaluating MLLMs. The new benchmark incorporates the tasks of object counting and multi-class identification for fine-grained perceptual abilities.
* The authors tried a wide range of visual tokenizers: DeiT, CLIP, MAE, DINO, analyzed their behaviors on the datasets, and reached several insights, including (1) supervision leads to a better understanding of semantics; (2) patch-level supervision improves fine-grained perceptual abilities.
* Based on the above insights, the authors combined the best of both worlds by introducing "feature distillation" to apply patch-level supervision to pretrained models. This improves the fine-grained perceptual abilities significantly.

**Strengths:**

This paper shows the following strengths:

1. The authors have conveyed their design choices and insights via extensive experiments. For example, the insights of the visual tokenizers are derived from the experiments of various models and datasets.
2. The benchmark constructed could be useful for future studies on multi-modal models with LLMs.
3. The final approach of feature distillation is straightforward and clearly improves the performance on the additional datasets of object counting and multi-class identification.

**Weaknesses:**

I think the authors may want to pay attention to the following weaknesses. I will include the specific questions in the next section.
1. I think the authors need to **better define the important notions** and consequently **clarify the difference to previous works**.
* Describing the difference between "overall semantics" and "fine-grained perception" is especially critical. Both terms seem vague to the readers but are the foundation of the evaluation benchmarks and insights in the paper.
* I understand the intuition of adding object counting and multi-class identification. However, their separation from VQA is unclear. I suggest improving the current clarification in Sec. 2.1 according to my questions below.
2. The method of this paper is quite simple (in a good way) by combining the techniques of feature distillation with patch-level supervision. However, I found the following things need improvement:
* Writing. Since the method is simple, it is always better to clearly explain the method. Sadly, I could hardly find a grounded description of the method in both Sec. 2.1 and Sec. 3.2. Adding proper equations and detailed clarifications will help the reader to fully understand the approach.
3. I personally think that the title of the paper is a little **overclaiming**. I will give two perspectives:
* I know that it is hard to define "good," but what the paper achieves is only **making MLLMs better at object counting multi-class identification** using patch-level feature distillation, without improving the overall semantic understanding. So concentrating on what is achieved is more precise, in my own opinion.
* When we talk about visual tokenizers, a lot of variables also need to be covered, such as the architecture, training techniques, etc. The paper mainly focuses on the types of supervision without investigating the other variables, which might not be sufficient for a grand title or claim. For example, will switching the claim of the paper to concentrate on "supervision" be better?

**Questions:**

Here are my questions. The authors are also welcome to reply to the points raised in the "Weaknesses" section here.

1. **Difference between VQA and the additional introduced tasks.** I think this is my most important concern of the paper. Specifically, I am not convinced by the author's explanation in Sec. 2.1 and would like more details. According to my observation on VQAv2, there are plenty of questions asking about the *number of objects* and *small objects in the image*. In my opinion, these are already sufficient for fine-grained perception. As the evaluation is the foundation for the points in this paper, I have the following questions:
* A clearer definition of "overall semantics" and "fine-grained perception" with grounded examples and statistics.
* Quantitatively compare VQAv2, Image captioning, and the datasets used for object counting and multi-class identification to show how the new tasks truly correspond to what this paper claims.
2. **Is semantic understanding and fine-grained understanding really the key?** Echoing the question above, I notice that the additional datasets rely on MS-COCO as their basis, which naturally has a distribution shift. I acknowledge that the different performances reflected in these datasets are intriguing, but I want quantitative evidence or a reasonable explanation attributing the reason to fine-grained visual understanding.
3. The part related to masked image modeling confuses me. In Sec. 2.2, the authors discovered that MAE is particularly better at fine-grained visual understanding, but found that masked image modeling hurts the performance in Sec. 3.1 and Table 3. These claims are conflicting and need better clarification. For example, the authors mentioned the train-test difference in introducing the mask tokens: (a) are there any ways to quantify this? (b) Does MAE exhibit the same issue? (c) If using a mask token can be made further tuning to function properly, does the claim "masked pertaining may not lead to ..." need to be revised?
4. I expect **more informative qualitative results**. I appreciate the authors picking qualitative comparison with previous methods (e.g. BLIP-2, in this case). However, these visualizations do not provide sufficient insights into *how the fine-grained abilities are improved?* In the worst case, the readers would consider cherry-picking or writing a program to filter several examples that your method outperforms another approach straightforward. Therefore, I have a suggestion/question:
* Is it doable to find some ways to visualize the attention/feature activation in such vision-language models? For example, if the feature activation in the 5-th example "dies bike exists in the image" shows better focus on the bike for your GVT model, it would be really convincing. Some papers have done similar analysis for your reference ([Shi et al, Fig. 1 and Fig. 6; Pang et al., Fig. B]), but your case might be complicated by the perceiver architecture. Finally, having such qualitative results will greatly add to your simple method and better convince me.

Shi et al. Top-Down Visual Attention from Analysis by Synthesis
Pang et al. Frozen Transformers in Language Models Are Effective Visual Encoder Layers

---

> ### Author Response · Authors · 2023-11-21
>
> We sincerely thank the reviewer for appreciating the insights, the usefulness of the conclusions, and the methodological contribution of our paper. We have revised our paper according to your constructive feedback (Sec 2.1, 3.1, 3.2, 4.4, Figure 1, 4, 5). Our responses are as follows:
>
>
> **Weaknesses 1.1 & Question 1.1 [semantics & fine-grained perception]**
>
> We used “overall semantics” to represent “global understanding” of the image while using “fine-grained perception” to signify local or regional understanding of the image. Based on your suggestions, we have revised our explanation for this part. Please see below.
>
> **Weaknesses 1.2 & Question 1.2 [Comparing OC&MCI with VQA]**
>
> We agree that there are questions asking "how many" in VQAv2. However, these questions are sometimes coupled with other semantic understanding. For example, VQAv2 may ask questions like "How many people are boarding a bus?". To answer this question, the model needs to understand semantic relations. In contrast, our OC and MCI questions are decoupled from such semantics, making them more focused on fine-grained visual understanding.
>
>
> To better clarify the difference in these tasks, we revised  Figure 1, Sec 2.1 and Table 1 in our revision, providing better distinguish and comparison among the four tasks. In particular, we revise the evaluation dimension of VQAv2 as "general semantics" due to its versatility to evaluate various visual understandings such as relations and actions. We take image captioning as the task to evaluate the overall semantics, which requires the model to understand the global information of the image. Then, OC and MCI are responsible for evaluating region-level and instance-level understanding, respectively.
>
> **Weaknesses 2 [Refining Sec 2.1 and Sec 3.2]**
>
> Thank you for your suggestion, we have revised the paper accordingly. Please refer to Section 2.1 (Page 3-4)， Sec 3.1 & 3.2 (Pages 5-6) of the updated paper.
>
>
> **Weaknesses 3 [Paper Title]**
>
> Thank you for your suggestions, we have revised our title to "What makes for good supervision for visual tokenizers in LLM?" We have also updated our writing in the revision.
>
> **Question 2 [Fine-grained visual understanding]**
> We provide a comparison on the SEED-Bench [4], which is a recently released MLLM benchmark and built upon the images in CC3M . The results show that FD can still improve the performance on this web image benchmark across a wider range of tasks.
>
> |			 |Scene Understanding|Instance Identity|Instance Location|Instance Attributes|Instance Counting|Spatial Relations|Instance Interactions|Visual Reasoning|Avg|
> |---|---|---|---|---|---|---|---|---|---|
> |GVT wo FD|41.26|34.30|31.40|**29.84**|34.81|**32.98**|**31.96**|50.75|35.93|
> |GVT	      |**41.74**|**35.50**|**31.79**|29.45|**36.17**|31.96|**31.96**|**51.06**|**36.20**|
>
> Notably, FD improves performance more on fine-grained understanding tasks such as instance identity, location and counting.
>
> **Question 3**
>
> Thank you for your question. We would like to clarify that the difference comes from not only the pre-training strategy, but also from tuning or freezing the visual tokenizer. For example, Sec 2.2 shows **fine-tuned** MAE is better at fine-grained visual understanding while Sec 3.1 shows **frozen** Masked-Image Modeling visual tokenizer (EVA) is unsatisfactory.
>
> a.We use the pretrained and ImageNet-1k fine-tuned models (MAE and EVA) to extract features of images in the validation set of ImageNet, and compare the cosine similarity between features and average them across the classes. The similarities of MAE and EVA are 0.76 and 0.71, respectively. The results show that there is indeed a distribution difference between features extracted by pretrained and fine-tuned models, demonstrating the existence of train-test mismatch.
>
> b.Yes, there is a similar train-test mismatch in MAE, In MAE pretraining, even though the [MASK] token is not input into the encoder, only partial image patches are fed into it.  In Table 2, Fine-tuned MAE is significantly better than Frozen MAE, suggesting a similar train-test mismatch undermines the performance.
>
> c.Thank you for your suggestion, we have revised our statement in Abstract (Page 1), Sec 3.1 (Page 6) and Sec 6 (page 9) to make it more rigorous.
>
>
> **Question 4**
>
> Thank you for your suggestions. Following [1], we visualized the attention map of the visual tokenizer in Figure 4  in our revision. The results demonstrate the attention after FD focuses more on fine-grained details while exhibiting higher diversity.
>
>
> **Reference**
>
> [1] Pang et al. Frozen Transformers in Language Models Are Effective Visual Encoder Layers
>
> Thank you again for appreciating our work, and we would be grateful if you could raise the score.
> Please let us know if there is anything we can do to convince you to further raise the score.

---

> > ### Comment · Reviewer_NrqM · 2023-11-22
> >
> > Thank you for your revisions and response!
> >
> > Overall, I think your response addresses my questions and concerns, e.g., a clearer clarification for VQAv2, discrepancies of MAE, and additional visualizations.
> >
> > Some follow-ups:
> > - The new Figure 3 provides more insight and better convinces the readers. I suggest adding the question (in Figure 4) at the bottom of Figure 3 to provide more context so that people better understand why the model focuses on broccoli.
> > - Your response of "fine-grained visual understanding" leverages another benchmark to illustrate the improvement in fine-grained perception. I highly encourage you to include this in your paper (it seems not in the manuscript now?)

---

> > > ### Author Response · Authors · 2023-11-23
> > >
> > > Thank you for your reply! The authors are delighted to address your concerns.
> > >
> > > For the follow-ups, we have updated our paper according to your suggestions:
> > >
> > > * Thank you for the suggestion, we added the question to Figure 3 for better visualization.
> > >
> > > * We include these comparisons (Table 7) in the revision.
> > >
> > > Thank you again for your response!

---

### Official Review · Reviewer_v493 · 2023-10-31

**Soundness:** 3 good
**Presentation:** 3 good
**Contribution:** 2 fair
**Rating:** 5
**Confidence:** 5

**Summary:**

This paper empirically investigates different visual tokenizers that can be used for multimodal large language models. Based on a set of experiments, authors utilize a visual tokenizer pretrained on large-scale datasets and properly integrate it with patch-level supervision. Extensive experiments on the new benchmark (GVTBench) demonstrate the effectiveness of the proposed model.

**Strengths:**

1. The analysis of different visual tokenizers (e.g., Table 1) is very informative.

2. The writing is clear and easy to follow.

3. Compared to the baselines, there are consistent improvements on OC and MCI tasks.

**Weaknesses:**

1. The main concern is about the experimental setup. Besides CC3M and SBU, authors use Visual Genome, MS-COCO, Object365 and OpenImages V6 to train the visual tokenizer. Based on the descriptions in Section 2.1, questions in OC and MCI tasks are object-centric. Such pretraining datasets provide a privilege for the proposed model. As shown in table 4, most of the improvements come from OC and MCI.

2. In table 1 and 2, authors use CLIP for ablation study but switch to EVA-CLIP for the remaining experiments. Why is there a inconsistency?

3. Based on table 5 and 6, it seems that there are consistent improvements by explicitly utilizing object-level annotations during pretraining.  This does not necessarily mean that the proposed tokenizer is a good visual tokenizer, especially considering the performance drop on VQA and image captioning tasks.

**Questions:**

See the weakness. The main concern is that the pretraining image-text pairs contain a significant amount of object-level annotations. This leads an unfair comparison when the downstream tasks are most object-centric questions.

---

> ### Author Response · Authors · 2023-11-21
>
> We sincerely thank the reviewer for appreciating the empirical study, writing, and performance improvement of our work. We have revised the paper according to your insightful comments (Sec 2.1, 3.1, 3.2, 4.4, Figure 1, 4, 5). Our responses are as follows:
>
> **Weaknesses 1 [Experimental Setup]**
>
> We would like to highlight that this work desires to provide a useful empirical study on what makes for good visual tokenizers for multimodal LLMs, which can understand both semantic and fine-grained visual information. We use these datasets (especially Object365 and OpenImages V6) for instruction-tuning to teach the model to better understand fine-grained visual information, so that the distinct fine-grained understanding capability in different visual tokenizers can be amplified in our evaluation, making it more straightforward to quantitatively compare the different visual tokenizers.
>
> We agree that our work utilizes different training protocols with previous models (e.g. MiniGPT-4). In fact, the training protocols in previous models are generally different, making it extremely difficult to compare these models in a unified setting. As such, our results in Table 4 aim to provide a system-level comparison, so as to demonstrate the superiority of our model. We would like to highlight that for other experiments (empirical studies and ablation studies), we strictly adopted identical settings, so as to safely provide trustworthy conclusions.
>
> In summary, we would like to clarify that, instead of achieving SOTA state-of-the-art performance given a fixed amount of data, a more important goal of our work is to study visual tokenizers with different pretraining strategies, so as to provide insight on constructing effective visual tokenizers for future researchers in the community.
>
>
> **Weaknesses 2 [CLIP v.s. EVA-CLIP]**
>
> We use CLIP for empirical studies since it is a prime model and widely used in other tasks.
> To build a strong MLLM  for the other experiments, we resort to EVA-CLIP for a stronger model.
>
> We provide the following results to show that EVA-CLIP is indeed stronger than CLIP in our GVTBench, which is aligned with the observations in [1].
> |VQA|Caption|COCO-OC|COCO-MCI|VCR-OC|VCR-MCI|Avg|
> |---|---|---|---|---|---|---|
> |CLIP|47.7|64.2|45.5|**88.0**|34.5|68.8|58.1|
> |EVA-CLIP|**48.1**|**65.3**|**46.0**|**88.0**|**34.6**|**69.1**|**58.5**|
>
> **Weaknesses 3 [Comparison in Table 5 & 6]**
>
> We agree that using different levels of annotations may lead to distinct final performances. For the comparisons in Table 5 and 6, we strictly used identical settings and show that FD indeed improves fine-grained visual understanding. From such fair comparisons, we safely conclude that the proposed GVT is a better visual tokenizer.
>
> **Reference**
>
> [1] Sun et al. http://arxiv.org/abs/2303.15389. Arxiv2303.15389
>
>
> Thank you again for appreciating our work, and we would be grateful if you could raise the score. Please let us know if there is anything we can do to convince you to further raise the score.

---

### Official Review · Reviewer_p5cW · 2023-11-02

**Soundness:** 2 fair
**Presentation:** 3 good
**Contribution:** 2 fair
**Rating:** 5
**Confidence:** 4

**Summary:**

This work studies an intriguing problem about how the visual tokenizer affects the final performance of LMMs. The authors started with a comprehensive evaluation on various vision encoders, obtained from supervised learning, self-supervised, and weakly-supervised learning. The results demonstrate that a visual tokenizer that possesses both semantic and fine-grained representations achieves the best performance. The author also studied some fine-grained CLIP models and found that it usually tend to loss generic semantic representations due to task overfitting. In the end, the authors proposed their own visual tokenizer training strategy by distilling the CLIP knowledge from one model to another. It turns out that such a simple distillation strategy can preserve a good quality of semantics while enhancing the spatial representations as well. The experimental results on the proposed GVTBench and other question-answering tasks show the superiority to previous models in general.

**Strengths:**

1. I appreciate the authors for conducting a good experimental study on the effect of different visual tokenizers for large multimodal models. It has been a routine that almost all LMMs use CLIP vision encoder as the visual tokenizer. Few people pay enough attention to the vision encoder itself, though it is a very important factor in LMMs.

2. The study on different visual tokenizers suggests that CLIP is still one of the best visual tokenizers for LMMs, despite it is not necessarily able to provide fine-grained representations. This is somehow surprising but also expected because all of the other used visual tokenizers were trained either with much fewer image-text pairs or purely image data at similar scales.

3. The authors further proposed a new benchmark and a simple distillation method to investigate and enhance the fine-grained visual understanding capability for LMMs. The experimental results demonstrate the superiority of the proposed method to previous ones.

**Weaknesses:**

1. As mentioned earlier, I like the first part of this work studying the effectiveness of different visual tokenizers for LMMs. The argument of LMMs needing both semantic-rich and fine-grained representations is valid and shared by many but has never been demonstrated experimentally before. However, I do not see a clear motivation to propose a distillation method to address the problem derived from the observation at the first part. The authors simply distill the visual representations from EVA-CLIP to another randomly initialized vision encoder and presume that the new model has better semantic and fine-grained representations.

2. I do not buy that the proposed method learns better fine-grained representations in that the input images for distillation are still 224x224, and the distillation is simply copying one network to another based on a limited number of images from ImageNet-1K. Despite the better performance in Table 4, I can see the unfairness when comparing the proposed method with previous ones -- e.g. different models have different settings and pretraining data. In the ablation study, the authors did show some improvements after distillation on the new benchmarks. Two question marks arise. First, why the authors use EVA-CLIP which proved to be defective in Table 3. Second, is it because the distillation pulls the feature space closer to the target domain, e.g., web images->imagenet->COCO? In Table 6, the gap actually is very marginal (seems the averaged gap does not match the individual gaps for Flant5-xxl).

3. The experimental study is not enough to demonstrate the effectiveness of the distilled visual tokenizer. The authors should examine the visual tokenizers in a wider range of vision and vision-language tasks. That being said, the authors attempt to find a more generic visual tokenizer that can provide semantic-rich and fine-grained representations but fail to demonstrate the generality of the proposed visual tokenizer by itself.

**Questions:**

1. I think the authors should explain a bit more about how to derive the proposed distillation methods to improve the visual tokenizer and why the new tokenizer is supposed to learn more fine-grained representations.

2. The empirical study on the effectiveness of the proposed visual tokenizer is insufficient, not to mention some unfairness in the comparisons with previous works. To solidly prove the generality of the proposed distillation technique, the authors should provide more benchmarking results across a wide range of tasks. I doubt distilling on ImageNet-1k is enough to achieve the goal.

3. Why the authors did not use CLIP released by OpenAI as the teacher, but rather used EVA-CLIP as the teacher model, despite knowing that it is inferior to CLIP and other models?

---

> ### Author Response · Authors · 2023-11-21
>
> The authors would like to thank you for your appreciation of the empirical study, conclusions, and superior performance of our work. We have revised our paper according to your valuable suggestions (Sec 2.1, 3.1, 3.2, 4.4, Figure 1, 4, 5). Ours responses are as follows.
>
> **Weaknesses 1 & Question 1 [Motivation of our method]**
>
> * We observe from Table1-3 that CLIP is strong at semantic understanding (VQA and image captioning) while inferior to MAE at fine-grained understanding (OC and MCI); MAE is superior in fine-grained visual understanding compared to other self-training models, which can be attributed to its region/patch-level pretraining.
>
> * Considering that MAE's ability of fine-grained visual understanding arises from its patch-level pretraining (which is realized by the patch reconstruction), we therefore seek to enhance CLIP with patch-level training. Since the mask-based patch-level training (EVA) is proven to be unsatisfactory (Table 3), distilling features matches our demands exactly because we can use both CLIP and patch-level distillation target.
>
> **Weaknesses 2.1 & Question 3 [Why prefer EVA-CLIP over EVA]**
>
> We would like to clarify that EVA [1] and EVA-CLIP [2] are two different models. EVA is a Mask-Image-Modeling pretrained model which uses CLIP feature as region supervision, while EVA-CLIP is a contrastive vision-language pretrained model. In essence, EVA is more of an MAE model while EVA-CLIP is more of a CLIP model. Table 3 shows that **EVA** is less useful but we use **EVA-CLIP** as a stronger version of CLIP for feature distillation to achieve a strong MLLM. The comparison between CLIP and EVA-CLIP is shown as follows:
>
> ||VQA|Caption|COCO-OC|COCO-MCI|VCR-OC|VCR-MCI|Avg|
> |---|---|---|---|---|---|---|---|
> |CLIP|47.7|64.2|45.5|**88.0**|34.5|68.8|58.1|
> |EVA-CLIP|**48.1**|**65.3**|**46.0**|**88.0**|**34.6**|**69.1**|**58.5**|
>
> The results show that EVA-CLIP is generally better than CLIP in various tasks.
>
>
> **Weaknesses 2.2  & Question 2 & Weaknessnes 3 [Why FD on imagenet][More benchmarking results][Generallity of visual tokenizer]**
>
> **[Why FD on imagenet]** The reason that we chose ImageNet as the dataset for distillation is due to its high quality of images. Furthermore, we found that the performance improvements are generalizable across various different datasets (See the point below).
>
> **[More benchmarking results and generality of visual tokenizer]**
> We provide the evaluation comparison on the SEED-Bench [4], which is a recently released MLLM benchmark and built upon web-images in CC3M . The results show that FD can still improve the performance on the web image benchmark across a wider range of tasks, especially on fine-grained understanding tasks such as instance identity, location, and counting.
>
> |			 |Scene Understanding|Instance Identity|Instance Location|Instance Attributes|Instance Counting|Spatial Relations|Instance Interactions|Visual Reasoning|Avg|
> |---|---|---|---|---|---|---|---|---|---|
> |GVT wo FD|41.26|34.30|31.40|**29.84**|34.81|**32.98**|**31.96**|50.75|35.93|
> |GVT	      |**41.74**|**35.50**|**31.79**|29.45|**36.17**|31.96|**31.96**|**51.06**|**36.20**|
>
> These results show that our visual tokenizer is effective at various vision-language tasks. On the other hand, the effectiveness of FD on various vision tasks has been shown in [3].
>
>
> **Question 2 [Unfair comparison with Previous work]**
>
> We agree that previous work (MiniGPT-4, LLaVa and BLIP-2) utilize different training protocols. However, it is difficult to complemently align all the settings due to the distinct differences among these models. Our results in Table 4 (Table 5 in ours revision) aims to provide a systematic comparison with existing models, so as to demonstrate our model is able to achieve superior performance.
> However, we would like to highlight that, in our empirical studies and ablation studies, the comparisons follow the same experimental setting. The fair comparison in these experiments enable us to draw useful conclusions and reliable insights on the training supervision of visual tokenizers.
>
> **Weaknesses 2.3 [Gaps in Table 6]**
>
> We apologize for wrongly computing the average in Table 6 and have updated it in revision.
>
>
>
> **Reference**
>
> [1] Fang et al. Exploring the Limits of Masked Visual Representation Learning at Scale. In CVPR 2023
>
> [2] Sun et al. http://arxiv.org/abs/2303.15389. Arxiv2303.15389
>
> [3] Wei et al. Contrastive Learning Rivals Masked Image Modeling in Fine-tuning via Feature Distillation. Arxiv 2205.14141
>
> [4] Li et al. SEED-Bench： Benchmarking Multimodal LLMs with Generative Comprehension. Arxiv2307.16125
>
>
> Thank you again for appreciating our work, and we would be grateful if you could raise the score. Please let us know if there is anything we can do to convince you to further raise the score.

---

### Meta-Review · Area_Chair_sT5v · 2023-12-12

**Metareview:**

The paper received mixed scores — two borderline rejects and one borderline accept. As a strength of this work, and noted by the reviewers, the problem of evaluating the effectiveness of different visual tokenizers for LLMs is interesting. At the same time, concerns were raised about unclear motivation, unfair comparisons, and insufficient experiments to evaluate the generalization of the proposed distillation-based method. Despite the effort made by the authors during the rebuttal period, and the additional experiments on seed-bench, the AC considers the following remain major concerns. As pointed out by reviewer v493, the use of data tailored to object-centric tasks does seem to provide an advantage to the proposed method, making table 5 in the revised version misleading. While other experiments for assessing pre-training methods for visual tokenizers are fair, most of the conclusions (e.g., regarding tuning the visual tokenizer and patch-level supervision for capturing fine-grained details) are expected. In addition, as pointed out by the reviewers, for an empirical study more experiments would be important (also to better support the effectiveness of the proposed GVT visual tokenizer). Based on these reasons, the AC recommends rejection, but encourages the authors to further improve the paper for another top conference or journal submission.

**Justification For Why Not Higher Score:**

Some of the results in the paper are misleading, as independently pointed out by the reviewers. In addition, for an empirical study a more comprehensive experimental analysis would be important to support the claims in the paper.

**Justification For Why Not Lower Score:**

N/A

---

### Decision · Program_Chairs · 2024-01-16

Reject